# A Game Theoretic Approach to Class-wise Selective Rationalization

**Shiyu Chang**[1,2*]    **Yang Zhang**[1,2*]    **Mo Yu**[2*]    **Tommi S. Jaakkola**[3]
[1]MIT-IBM Watson AI Lab    [2]IBM Research    [3]CSAIL MIT
{shiyu.chang,yang.zhang2}@ibm.com   yum@us.ibm.com   tommi@csail.mit.edu

## Abstract

Selection of input features such as relevant pieces of text has become a common technique of highlighting how complex neural predictors operate. The selection can be optimized post-hoc for trained models or incorporated directly into the method itself (self-explaining). However, an overall selection does not properly capture the multi-faceted nature of useful rationales such as pros and cons for decisions. To this end, we propose a new game theoretic approach to class-dependent rationalization, where the method is specifically trained to highlight evidence supporting alternative conclusions. Each class involves three players set up competitively to find evidence for factual and counterfactual scenarios. We show theoretically in a simplified scenario how the game drives the solution towards meaningful class-dependent rationales. We evaluate the method in single- and multi-aspect sentiment classification tasks and demonstrate that the proposed method is able to identify both factual (justifying the ground truth label) and counterfactual (countering the ground truth label) rationales consistent with human rationalization. The code for our method is publicly available[2].

## 1   Introduction

Interpretability is rapidly rising alongside performance as a key operational characteristics across NLP and other applications. Perhaps the most straightforward means of highlighting how a complex method works is by selecting input features relevant for the prediction (e.g., [19]). If the selected subset is short and concise (for text), it can potentially be understood and verified against domain knowledge. The selection of features can be optimized to explain already trained models [24], incorporated directly into the method itself as in self-explaining models [19, 12], or optimized to mimic available human rationales [8].

One of the key questions motivating our work is extending how rationales are defined and estimated. The common paradigm to date is to make an *overall* selection of a feature subset that maximally explains the target output/decision. For example, maximum mutual information criterion [12, 19] chooses an overall subset of features such that the mutual information between the feature subset and the target output decision is maximized, or, equivalently, the entropy of the target output decision conditional on this subset is minimized. Rationales can be multi-faceted, however, involving support for different outcomes, just with different degrees. For example, we could understand the overall sentiment associated with a product in terms of weighing associated pros and cons contained in the review. Existing rationalization techniques strive for a single overall selection, therefore lumping together the facets supporting different outcomes.

We propose the notion of *class-wise rationales*, which is defined as multiple sets of rationales that respectively explain support for different output classes (or decisions). Unlike conventional

---

[*]Authors contributed equally to this paper.
[2]https://github.com/code-terminator/classwise_rationale

rationalization schemes, class-wise rationalization takes a candidate outcome as input, which can be different from the ground-truth class labels, and uncovers rationales specifically for the given class. To find such rationales, we introduce a game theoretic algorithm, called *Class-wise Adversarial Rationalization* (CAR). CAR consists of three types of players: factual rationale generators, which generate rationales that are consistent with the actual label, counterfactual rationale generators, which generate rationales that counter the actual label, and discriminators, which discriminate between factual and counterfactual rationales. Both factual and counterfactual rationale generators try to competitively "convince" the discriminator that they are factual, resulting in an adversarial game between the counterfactual generators and the other two types of players.

We will show in a simplified scenario how CAR game drives towards meaningful class-wise rationalization, under an information-theoretic metric, which is a class-wise generalization of the maximum mutual information criterion. Moreover, empirical evaluation on both single- and multi-aspect sentiment classification show that CAR can successfully find class-wise rationales that align well with human understanding. The data and code will become publicly available.

## 2   Related Work

There are two lines of research on generating interpretable features of neural network. The first is to directly incorporate the interpretations into the models, *a.k.a* self-explaining models [3, 4, 5, 15]. The other line is to generate interpretations in a post-hoc manner. There are several ways to perform post-hoc interpretations. The first class of method is to explicitly introduce a generator that learns to select important subsets of inputs as explanations [12, 19, 21, 30, 31], which often comes with some information-theoretic properties. The second class is to evaluate the importance of each input feature via backpropagation of the prediction. Many of these methods utilize gradient information [6, 20, 25, 26, 27, 28], while techniques like local perturbations [11, 13, 16, 22] and Parzen window [7] have also been used to loose the requirement of differentiability. Finally, the third class is locally fitting a deep network with interpretable models, such as linear models [2, 24]. There are also some recent works trying to improve the fidelity and/or stability of post hoc explanations by including the explanation mechanism in the training procedure [17, 18].

Although none of the aforementioned approaches can perform class-wise rationalization, gradient-based methods can be intuitively adapted for this purpose, which produces explanations toward a certain class by probing the importance with respect to the corresponding class logit. However, as noted in [24], when the input feature is far away from the corresponding class, the local gradient or perturbation probe can be very inaccurate. Evaluation of such methods will be provided in section 5.

## 3   Class-wise Rationalization

In this section, we will introduce our adversarial approach to class-wise rationalization. For notations, upper-cased letters, *e.g.* $X$ or $\boldsymbol{X}$, denote random variables or random vectors respectively; lower-cased letters, *e.g.* $x$ or $\boldsymbol{x}$, denote deterministic scalars or vectors respectively; script letters, *e.g.* $\mathcal{X}$, denote sets. $p_{X|Y}(x|y)$ denotes the probability of $X = x$ conditional on $Y = y$. $\mathbb{E}[X]$ denotes expectation.

### 3.1   Problem Formulation

Consider a text classification problem, where $\boldsymbol{X}$ is a random vector representing a string of text, and $\boldsymbol{Y} \in \mathcal{Y}$ represents the class that $\boldsymbol{X}$ is in. The class-wise rationalization problem can be formulated as follows. For any input $\boldsymbol{X}$, our goal is to derive a class-wise rationale $\boldsymbol{Z}(t)$ for any $t \in \mathcal{Y}$ such that $\boldsymbol{Z}(t)$ provides evidence supporting class $t$. Each rationale can be understood as a masked version $\boldsymbol{X}$, *i.e.* $\boldsymbol{X}$ with a subset of its words masked away by a special value (*e.g.* 0). Note that class-wise rationales are defined for *every* class $t \in \mathcal{Y}$. For $t = Y$ (the correct class) the corresponding rationale is called factual; for $t \neq Y$ we call them counterfactual rationales. For simplicity, we will focus on two-class classification problems ($\mathcal{Y} = \{0, 1\}$) for the remainder of this section. Generalization to multiple classes will be discussed in appendix A.4.

As a clarification, notice that during inference, the class $t$ that is provided to the system does *not* need to be the ground truth. No matter what $t$ is provided, factual or counterfactual, the algorithm is supposed to try its best to find evidence in support of $t$. Therefore, the inference does not need to

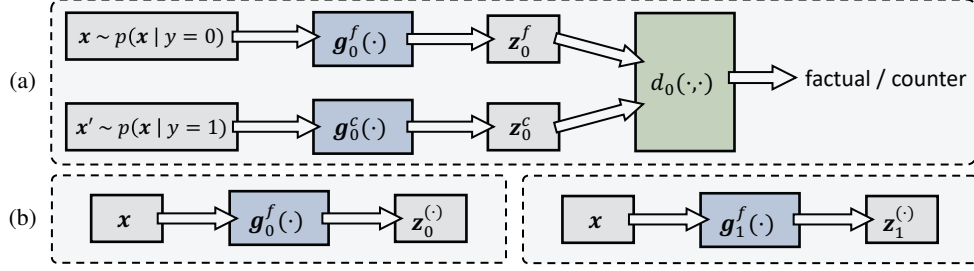

Figure 1: CAR training and inference procedures of the class-0 case. (a) The training procedure. (b) During inference, there is no ground truth label. In this case, we will always trigger the factual generators.

access the ground truth label. However, the training of the algorithm requires the ground truth label $Y$, because it needs to learn the phrases and sentences that are informative of each class.

## 3.2 The CAR Framework

CAR uncovers class-wise rationales using adversarial learning, inspired by outlining pros and cons for decisions. Specifically, there are two *factual rationale generators*, $g_t^f(X)$, $t \in \{0, 1\}$, which generate rationales that justify class $t$ when the actual label agrees with $t$, and two *counterfactual rationale generators*, $g_t^c(X)$, $t \in \{0, 1\}$, which generate rationales for the label other than the ground truth. Finally, we introduce two discriminators $d_t(Z)$, $t \in \{0, 1\}$, which aim to discriminate between factual and counterfactual rationales, *i.e.*, between $g_t^f(X)$ and $g_t^c(X)$. We thus have six players, divided into two groups. The first group pertains to $t = 0$ and involves $g_0^f(X)$, $g_0^c(X)$ and $d_0(Z)$ as players. Both groups play a similar adversarial game, so we focus the discussion on the first group.

**Discriminator:** In our adversarial game, $d_0(\cdot)$ takes a rationale $Z$ generated by either $g_0^f(\cdot)$ or $g_0^c(\cdot)$ as input, and outputs the probability that $Z$ is generated by the factual generator $g_0^f(\cdot)$. The training target for $d_0(\cdot)$ is similar to the generative adversarial network (GAN) [14]:

$$d_0(\cdot) = \underset{d(\cdot)}{\operatorname{argmin}} -p_Y(0)\mathbb{E}[\log d(g_0^f(X))|Y = 0] - p_Y(1)\mathbb{E}[\log(1 - d(g_0^c(X)))|Y = 1]. \quad (1)$$

**Generators:** The factual generator $g_0^f(\cdot)$ is trained to generate rationales from text labeled $Y = 0$. The counterfactual generator $g_0^c(\cdot)$, in contrast, learns from text labeled $Y = 1$. Both generators try to convince the discriminator that they are factual generators for $Y = 0$.

$$g_0^f(\cdot) = \underset{g(\cdot)}{\operatorname{argmax}} \mathbb{E}[h_0(d_0(g(X)))|Y = 0], \quad \text{and} \quad g_0^c(\cdot) = \underset{g(\cdot)}{\operatorname{argmax}} \mathbb{E}[h_1(d_0(g(X)))|Y = 1],$$
$$\text{s.t.} \quad g_0^f(X)) \text{ and } g_0^c(X) \text{ satisfy some sparsity and continuity constraints.} \quad (2)$$

The constraints stipulate that the words selected as rationales should be a relatively small subset of the entire text (sparse) and they should constitute consecutive segments (continuous). We will keep the constraints abstract for generality for now. The actual form of the constraints will be specified in section 4. $h_0(\cdot)$ and $h_1(\cdot)$ are both monotonically-increasing functions that satisfy the following properties:

$$x h_0\left(\frac{x}{x + a}\right) \text{ is convex in } x, \quad \text{and} \quad x h_1\left(\frac{a}{x + a}\right) \text{ is concave in } x, \quad \forall x, a \in [0, 1]. \quad (3)$$

One valid choice is $h_0(x) = \log(x)$ and $h_1(x) = -\log(1 - x)$, which reduces the problem to the more canonical GAN-style problem. In practice, we find that other functional forms have more stable training behavior. As shown later, this generalization is closely related to $f$-divergence.

Figure 1(a) summarizes the training procedure of these three players. As can be seen, $g_0^c(\cdot)$ plays an adversarial game with both $d_0(\cdot)$ and $g_0^f(\cdot)$, because it tries to trick $d_0(\cdot)$ into misclassifying its output as factual, whereas $g_0^f(\cdot)$ helps $d_0(\cdot)$ make the correct decision. The other group of players, $g_1^f(\cdot)$, $g_1^c(\cdot)$ and $d_1(\cdot)$, play a similar game. The only difference is that now the factual generator operates on text with label $Y = 1$, and the counterfactual generator on text with label $Y = 0$.

### 3.3 How Does It Work?

Consider a simple bag-of-word scenario, where the input text is regarded as a collection of words drawn from a vocabulary of size $N$. In this case, $\boldsymbol{X}$ can be formulated as an $N$-dimensional binary vector. $\boldsymbol{X}_i = 1$, if the $i$-th word is present, and $\boldsymbol{X}_i = 0$ otherwise. $p_{\boldsymbol{X}|Y}(\boldsymbol{x}|y)$ represents the probability distribution of $\boldsymbol{X}$ in natural text conditional on different classes $Y = y$.

The rationales $\boldsymbol{Z}_0^f$ and $\boldsymbol{Z}_0^c$ are also multivariate binary vectors. $\boldsymbol{Z}_{0,i}^f = 1$ if the $i$-th word is selected as part of the factual rationale, and $\boldsymbol{Z}_{0,i}^f = 0$ otherwise. $p_{\boldsymbol{Z}_0^f|Y}(\boldsymbol{z}|0)$ denotes the *induced* distribution of the factual rationales, which is only well-defined in the factual case ($Y = 0$). This distribution is determined by how $\boldsymbol{g}_0^f(\cdot)$ generates the rationales across examples. In the optimization problem, we will primarily make use of the induced distribution, and similarly for the counterfactual rationales.

To simplify our discussion, we assume that the dimensions of $\boldsymbol{X}$ are independent conditional on $Y$. Furthermore, we assume that the rationale selection scheme selects each word independently, so the induced distributions over $\boldsymbol{Z}_0^f$ and $\boldsymbol{Z}_0^c$ are also independent across dimensions, conditional on $Y$. Formally, $\forall \boldsymbol{x}, \boldsymbol{z} \in \{0,1\}^N, \forall y \in \{0,1\}$,

$$p_{\boldsymbol{X}|Y}(\boldsymbol{x}|y) = \prod_{i=1}^N p_{\boldsymbol{X}_i|Y}(\boldsymbol{x}_i|y), \;\; p_{\boldsymbol{Z}_0^f|Y}(\boldsymbol{z}|y) = \prod_{i=1}^N p_{\boldsymbol{Z}_{0,i}^f|Y}(\boldsymbol{z}_i|y), \;\; p_{\boldsymbol{Z}_0^c|Y}(\boldsymbol{z}|y) = \prod_{i=1}^N p_{\boldsymbol{Z}_{0,i}^c|Y}(\boldsymbol{z}_i|y). \quad (4)$$

Figure 2(left) plots $p_{\boldsymbol{X}_i|Y}(1|0)$ and $p_{\boldsymbol{X}_i|Y}(1|1)$ as functions of $i$ (the horizontal axis corresponds to sorted word identities). These two curves represent the occurrence of each word in the two classes. In the figure, the words to the left satisfy $p_{\boldsymbol{X}_i|Y}(1|0) > p_{\boldsymbol{X}_i|Y}(1|1)$, *i.e.* they occur more often in class 0 than in class 1. These words are most indicative of class 0, which we will call *class-0 words*. Similarly, the words to the right are called *class-1 words*.

Figure 2(left) also plots an example of $p_{\boldsymbol{Z}_{0,i}^f|Y}(1|0)$ and $p_{\boldsymbol{Z}_{0,i}^c|Y}(1|1)$ curves (solid, shaded curves), which represents the occurrence of each word in the factual and counterfactual rationales respectively. Note that these two curves must satisfy the following constraints:

$$p_{\boldsymbol{Z}_{0,i}^f|Y}(1|0) \le p_{\boldsymbol{X}_i|Y}(1|0), \quad \text{and} \;\; p_{\boldsymbol{Z}_{0,i}^c|Y}(1|1) \le p_{\boldsymbol{X}_i|Y}(1|1). \quad (5)$$

This is because a word can be chosen as a rationale *only if* it appears in a text, and this strict relation translates into an inequality constraint in terms of the induced distributions. As shown in figure 2(left), the $p_{\boldsymbol{Z}_{0,i}^f|Y}(1|0)$ and $p_{\boldsymbol{Z}_{0,i}^c|Y}(1|1)$ curves are always below the $p_{\boldsymbol{X}_i|Y}(1|0)$ and $p_{\boldsymbol{X}_i|Y}(1|1)$ curves respectively. For the remainder of this section, we will refer to $p_{\boldsymbol{X}_i|Y}(1|0)$ as the *factual upper-bound*, and $p_{\boldsymbol{X}_i|Y}(1|1)$ as the *counterfactual upper-bound*. What we intend to show is that the optimal strategy for both rationale generators in this adversarial game is to choose the class-0 words.

**The optimal strategy for the counterfactual generator:** We will first find out what is the optimal strategy for the counterfactual generator, or, equivalently, the optimal $p_{\boldsymbol{Z}_{0,i}^c|Y}(1|1)$ curve, given an arbitrary $p_{\boldsymbol{Z}_{0,i}^f|Y}(1|1)$ curve. The goal of the counterfactual generator is to fool the discriminator. Therefore, its optimal strategy is to match the the counterfactual rationale distribution with the factual rationale distribution. As shown in figure 2(middle), the $p_{\boldsymbol{Z}_{0,i}^c|Y}(1|1)$ (blue) curve tries to overlay with the $p_{\boldsymbol{Z}_{0,i}^f|Y}(1|1)$ (green) curve, within the limits of the counterfactual upper bound constraint.

**The optimal strategy for the factual generator:** The goal of the factual generator is to help the discriminator. Therefore, its optimal strategy, given the optimized counterfactual generator, is to "steer" the factual rationale distribution away from the counterfactual rationale distribution. Recall that the counterfactual rationale distribution always tries to match the factual rationale distribution, unless its upper-bound is binding. The factual generator will therefore choose the words whose factual upper-bound is much higher than the counterfactual upper-bound. These words are, by definition, most indicative of class 0. The counterfactual generator will also favor the same set of words, due to its incentive to match the distributions. Figure 2(right) illustrates the optimal strategy for the factual rationale under sparsity constraint

$$\sum_{i=1}^N \mathbb{E}[\boldsymbol{Z}_{0,i}^f] = \sum_{i=1}^N p_{\boldsymbol{Z}_{0,i}^f|Y}(1|1) \le \alpha. \quad (6)$$

The left-hand side in equation (6) represents the expected factual rationale length (in number of words). It also represents the area under the $p_{\boldsymbol{Z}_{0,i}^f|Y}(1|1)$ curve (the green shaded areas in figure 2).

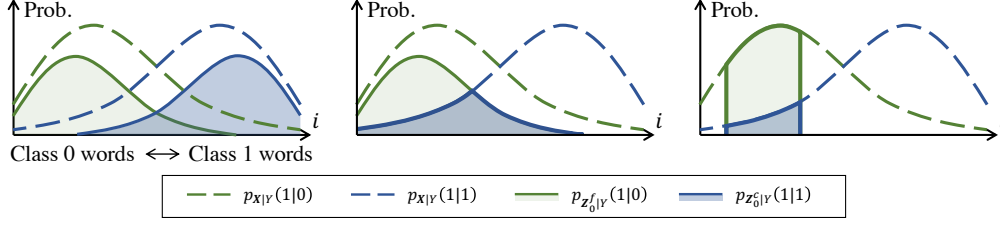

Figure 2: An illustration of how CAR works in the bag-of-word scenario with independence assumption (equation (4)). Left: example probability of occurrence of each word in the rationales from each class (solid lines), upper bounded by the probability of occurrence of each word in the natural text from each class (dashed lines). Middle: the optimal strategy for the counterfactual rationale is to match the factual rationale distribution, unless prohibited by the upper-bound. Right: the optimal strategy for the factual rationale is to steer away from the counterfactual rationale distribution, leveraging the upper-bound difference.

## 3.4 Information-theoretic Analysis

Now we are ready to embark on a more formal analysis of the effectiveness of the CAR framework, as stated in the following theorem.

**Theorem 1.** *In the bag-of-word scenario with the independence assumption as in equation* (4)*:*

**(1)** *Given the optimal $d_0(\cdot)$ and an arbitrary $\boldsymbol{g}_0^f(\cdot)$, the optimal $\boldsymbol{g}_0^c(\cdot)$ to equation* (2) *(left) will generate the counterfactual rationales that follow the following distribution:*

$$p_{\boldsymbol{Z}_{0,i}^c|Y}(1|1) = \min\left\{p_{\boldsymbol{Z}_{0,i}^f|Y}(1|0), p_{\boldsymbol{X}_i|Y}(1|1)\right\}. \tag{7}$$

**(2)** *Under some additional assumptions (see appendix A.1), given the optimal $d_0(\cdot)$ and the optimal $\boldsymbol{g}_0^c(\cdot)$, the optimal $\boldsymbol{g}_0^f(\cdot)$ to equation* (2) *(right) subject to the sparsity constraint as in equation* (6) *is given by $\boldsymbol{Z}_{0,i}^f = \boldsymbol{X}_{\mathcal{I}^*}$, where*

$$\mathcal{I}^* = \underset{\mathcal{I}}{\arg\max} \, \mathbb{E}_{\boldsymbol{X} \sim p_{\boldsymbol{X}|Y}(\cdot|0)} \left[ h\left( \frac{p_{\boldsymbol{X}_{\mathcal{I}}|Y}(\boldsymbol{X}_{\mathcal{I}}|0)}{p_{\boldsymbol{X}_{\mathcal{I}}}(\boldsymbol{X}_{\mathcal{I}})} \right) \right], \quad s.t. \quad p_{\boldsymbol{X}_i|Y}(1|0) > p_{\boldsymbol{X}_i|Y}(1|1), \forall i \in \mathcal{I}, \tag{8}$$

*where $\boldsymbol{X}_{\mathcal{I}}$ denotes a subvector of $\boldsymbol{X}$ containing $\boldsymbol{X}_i$, $\forall i \in \mathcal{I}$.*

The proof will be given in the appendix. To better understand equation (8), it is useful to first write down the mutual information between $\boldsymbol{X}_{\mathcal{I}}$ and $Y$, a similar quantity to which has been applied to the maximum mutual information criterion [12, 19].

$$I(Y; \boldsymbol{X}_{\mathcal{I}}) = \mathbb{E}_{\boldsymbol{X}, Y \sim p_{\boldsymbol{X}, Y}(\cdot, \cdot)} \left[ \log\left( \frac{p_{\boldsymbol{X}_{\mathcal{I}}|Y}(\boldsymbol{X}_{\mathcal{I}}|Y)}{p_{\boldsymbol{X}_{\mathcal{I}}}(\boldsymbol{X}_{\mathcal{I}})} \right) \right] = \sum_{y=0}^{1} p_Y(y) \mathbb{E}_{\boldsymbol{X} \sim p_{\boldsymbol{X}|Y}(\cdot|y)} \left[ \log\left( \frac{p_{\boldsymbol{X}_{\mathcal{I}}|Y}(\boldsymbol{X}_{\mathcal{I}}|y)}{p_{\boldsymbol{X}_{\mathcal{I}}}(\boldsymbol{X}_{\mathcal{I}})} \right) \right].$$
$$\tag{9}$$

As can be seen, there is a correspondence between equations (8) and (9). First, the $\log(\cdot)$ function in equation (9) is generalized a wider selection of functional forms, $h(\cdot)$. As will be shown in the appendix A.2, equation (8) applies the $f$-divergence [1], which is a generalization to the KL-divergence as applied in equation (9). Second, notice that equation (9) is decomposed into two class-dependent terms, while equation (8) is for class-0 generators only. It can be easily shown that the class-1 generators come with a similar theoretical guarantee that corresponds to the term with $y = 1$. Therefore, the target function in equation (8) can be considered as the component in the mutual information that is specifically related to class $0$. Hence we call it *class-wise mutual information*.

## 3.5 Coping with Degeneration

It has been pointed out in [32] that the existing generator-predictor framework in [12] and [19] can suffer from the problem of *degeneration*. Since the generator-predictor framework aims to maximize the predictive accuracy of the predictor, the generator and predictor can collude by selecting uninformative symbols to encode the class information, instead of selecting words and phrases that truly explain the class. For example, consider the following punctuation communication scheme: when $Y = 0$, the rationale would select only one comma ","; when $Y = 1$, the rationale would select only one period ".". This rationalization scheme guarantees a high predictive accuracy. However, this is apparently not what we expect. Such cases are called degeneration.

From section 3.3, we can conclude that CAR will not suffer from degeneration. This is because if the factual rationale generators attempt to select uninformative words or symbols like punctuation (*i.e.* words in the middle of the $x$-axis in figure 2), then the factual rationale distribution can be easily matched by the counterfactual rationale distribution. Therefore, this strategy is not optimal for the factual generators, whose goal is to avoid being matched by the counterfactual generators.

# 4 Architecture Design and Implementation

**Architecture with parameter sharing:** In our actual implementation, we impose parameter sharing among the players. This is motivated by our observation in sections 3.3 and 3.4 that both the factual and counterfactual generators adopt the same rationalization strategy upon reaching the equilibrium. Therefore, instead of having two separate networks for the two generators, we introduce one unified generator network for each class, a class-0 generator and a class-1 generator, with the ground truth label $Y$ as an additional input to identify between factual and counterfactual modes. Specifically, $g_0^c(\cdot)$ and $g_0^f(\cdot)$ now share the same parameters in a single generator network $g_0(\cdot, Y)$, where $g_0^f(\cdot) = g_0(\cdot, 0)$, and $g_0^c(\cdot) = g_0(\cdot, 1)$. Please note that after the parameter sharing, $g_0(\cdot, 0)$ and $g_0(\cdot, 1)$ are still considered as two distinct players, in the sense that they are still trained to optimize different target functions (equation (2)), and they still play the same adversarial game with each other. Similarly, $g_1^c(\cdot)$ and $g_1^f(\cdot)$ share the same parameters in a single generator network $g_1(\cdot, Y)$. We also impose parameter sharing between the two discriminators, $d_0(\cdot)$ and $d_1(\cdot)$, by introducing a unified discriminator, $d(\cdot, t)$, with an additional input $t$ to identify between the class-0 and class-1 cases. The trainable parameters are significantly reduced with parameter sharing.

Both the generators and the discriminators consist of a word embedding layer, a bi-direction LSTM layer followed by a linear projection layer. The generators generate the rationales by the independent selection process as proposed in [19]. At each word position $k$, the convolutional layer outputs a quantized binary mask $\boldsymbol{S}_k$, which equals to 1 if the $k$-th word is selected and 0 otherwise. The binary masks are multiplied with the corresponding words to produce the rationales. For the discriminators, the outputs of all the times are max-pooled to produce the factual/counterfactual decision.

For parameter sharing, we append the input class as a one-hot vector to each word embedding vector in both the generators and the discriminator. For the generators, the groundtruth class label $Y$ of each instance is appended; while for the discriminator, the class of generator $t$ used for generating the input rationale is appended.

**Training:** The training objectives are essentially equations (1) and (2). The only difference is that we instantiate the constraints in equation (2) transform it into a multiplier form. Specifically, the multiplier terms (or the regularization terms) are

$$\lambda_1 \left| \frac{1}{K} \mathbb{E}[\|\boldsymbol{S}\|_1] - \alpha \right| + \lambda_2 \mathbb{E}\left[ \sum_{t=2}^{K} |\boldsymbol{S}_k - \boldsymbol{S}_{k-1}| \right], \tag{10}$$

where $K$ denotes the number of words in the input text. The first term constrains on the sparsity of the rationale. It encourages that the percentage of the words being selected as rationales is close to a preset level $\alpha$. The second term constrains on the continuity of the rationale. $\lambda_1$, $\lambda_2$ and $\alpha$ are hyperparameters. The constraint is slightly different from the one in [19] in order have a more precise control of the sparsity level. The $h_0(\cdot)$ and $h_1(\cdot)$ functions in equation (2) are set to $h_0(\boldsymbol{x}) = h_1(\boldsymbol{x}) = \boldsymbol{x}$, which empirically shows good convergence performance, and which can be shown to satisfy equation (3). To resolve the non-differentiable quantization operation that produces $\boldsymbol{S}_t$, we apply the straight-through gradient computation technique [9]. The training scheme involves the following alternate stochastic gradient descent. First, the class-0 generator and the discriminator are updated jointly by passing one batch of data into the class-0 generator, and the resulting rationales, which contain both factual and counterfatual rationales depending on the actual class, are fed into the discriminator with $t = 0$. Then, the class-1 generator and the discriminator are updated jointly in a similar fashion with $t = 1$.

**Inference:** During the inference, the ground truth label is unavailable for fair comparisons with the baselines, therefore we have no oracle knowledge of which class is factual and which is counterfactual. In this case, we always trigger the factual generators, no matter what the ground truth is, as shown in figure 1(b). This is again justified by our observation in sections 3.3 and 3.4 that both the factual and counterfactual modes adopt the same rationalization strategy upon reaching the equilibrium. The

only reason why we favor the factual mode to the counterfactual mode is that the former has more exposure to the words it is supposed to select during training.

# 5 Experiments

## 5.1 Datasets

To evaluate both factual and counterfactual rationale generation, we consider the following three binary classification datasets. The first one is the single-aspect Amazon reviews [10] (book and electronic domains), where the input texts often contain evidence for both positive and negative sentiments. We use predefined rules to parse reviews containing comments on both the pros and cons of a product, which is further used for automatic evaluations. We also evaluate algorithms on the multi-aspect beer [23] and hotel reviews [29] that are commonly used in the field of rationalization [8, 19]. The labels of the beer review dataset are binarized, resulting in a harder rationalization task than in [19]. The multi-aspect review is considered as a more challenging task, where each review contains comments on different aspects. However, unlike the Amazon dataset, both beer and hotel datasets only contain factual annotations. The construction of evaluation tasks is detailed in appendix B.1.

## 5.2 Baselines

**RNP:** A generator-predictor framework proposed by Lei *et al.* [19] for rationalizing neural prediction (RNP). The generator selects text spans as rationales which are then fed to the predictor for label classification. The selection maximizes the predictive accuracy of the target output and is constrained to be sparse and continuous. RNP is only able to generate factual rationales.

**POST-EXP:** The post-explanation method generates rationales of both positive and negative classes based on a pre-trained predictor. Given the predictor trained on full-text inputs, we train two separate generators $g_0(X)$ and $g_1(X)$ on the data to be explained. $g_0(X)$ always generate rationales for the negative class and $g_1(X)$ always generate rationales for the positive class. The two generators are trained to maximize the respective logits of the fixed predictor subject to sparsity and continuity regularizations, which is closely related to gradient-based explanations [20].

To seek fair comparisons, the predictors of both RNP and POST-EXP and the discriminator of CAR are of the same architecture; the rationale generators in all three methods are of the same architecture. The hidden state size of all LSTMs is set to 100. In addition, the sparsity and continuity constraints are also in the same form as our method. It is important pointing out that CAR does not use any ground truth label for generating rationales, which follows the procedures discussed in section 4.

## 5.3 Experiment Settings

**Objective evaluation:** We compare the generated rationales with the human annotations and report the precision, recall and F1 score. To be consistent with previous studies [19], we evaluate different algorithms conditioned on a similar *actual* sparsity level in factual rationales. Specifically, the target factual sparsity level is set to around ($\pm 2\%$) 20% for the Amazon dataset and 10% for both beer and hotel review. The reported performances are based on the best performance of a set of hyperparameter values. For details of the setting, please refer to appendix B.2.

**Subjective evaluation:** We also conduct subjective evaluations via *Amazon Mechanical Turk*. Specifically, we reserve 100 randomly balanced examples from each dev set for the subjective evaluations. For the single-aspect dataset, the subject is presented with either the factual rationale or the counterfactual rationale of a text generated by one of the three methods (unselected words blocked). For the factual rationales, a success is credited when the subject correctly guess the ground-truth sentiment; for the counterfactual rationales, a success is credited when the subject is convinced to choose the opposite sentiment to the ground-truth. For the multi-aspect datasets, we introduce a much harder test. In addition to guessing the sentiment, the subject is also asked to guess what aspect the rationale is about. A success is credited only when both the intended sentiment *and* the correct aspect are chosen. Under this criterion, a generator that picks the sentiment words only will score poorly. We then compute the success rate as the performance metric. The test cases are randomly shuffled. The subjects have to meet certain English proficiency and are reminded that some of the generated

Table 1: Objective performances of selected rationales of the Amazon review dataset. The numbers in each column represent the sparsity level, precision, recall, and F1 score, respectively. Each domain is trained independently. All results are calculated in a "micro" perspective.

| Amazon | Book | | Electronic | |
|---|---|---|---|---|
| | Factual | Counterfactual | Factual | Counterfactual |
| RNP [19] | 18.6/55.1/20.1/29.5 | - | 20.7/49.7/22.8/31.3 | - |
| POST-EXP | 20.2/64.5/28.8/39.8 | 27.9/70.2/**35.8**/**47.4** | 18.6/64.1/27.8/38.8 | 15.3/72.6/**19.5**/**30.7** |
| CAR | 20.9/**68.7**/**31.9**/**43.6** | 15.2/**72.2**/20.2/31.5 | 21.2/**70.0**/**34.7**/**46.4** | 10.2/**76.4**/13.6/23.1 |

Table 2: Objective performances of selected factual rationales for both (a) beer and (b) hotel review datasets. Each aspect is trained independently. S, P, R, and F1 indicate the sparsity level, precision, recall, and F1 score.

|  | Beer | Appearance | | | | Aroma | | | | Palate | | | |
|---|---|---|---|---|---|---|---|---|---|---|---|---|---|
| | | S | P | R | F1 | S | P | R | F1 | S | P | R | F1 |
| (a) | RNP [19] | 11.9 | 72.0 | 46.1 | 56.2 | 10.7 | **70.5** | 48.3 | **57.3** | 10.0 | 53.1 | 42.8 | 47.5 |
| | POST-EXP | 11.9 | 64.2 | 41.4 | 50.4 | 10.3 | 50.0 | 33.1 | 39.8 | 10.0 | 33.0 | 26.5 | 29.4 |
| | CAR | 11.9 | **76.2** | **49.3** | **59.9** | 10.3 | 50.3 | 33.3 | 40.1 | 10.2 | **56.6** | **46.2** | **50.9** |

|  | Hotel | Location | | | | Service | | | | Cleanliness | | | |
|---|---|---|---|---|---|---|---|---|---|---|---|---|---|
| | | S | P | R | F1 | S | P | R | F1 | S | P | R | F1 |
| (b) | RNP [19] | 10.9 | 43.3 | 55.5 | 48.6 | 11.0 | 40.0 | 38.2 | 39.1 | 10.6 | 30.5 | **36.0** | 33.0 |
| | POST-EXP | 8.9 | 30.4 | 31.8 | 31.1 | 10.0 | 32.5 | 28.3 | 30.3 | 9.2 | 23.0 | 23.7 | 23.3 |
| | CAR | 10.6 | **46.6** | **58.1** | **51.7** | 11.7 | **40.7** | **41.4** | **41.1** | 9.9 | **32.3** | 35.7 | **33.9** |

rationales are intended to trick them via word selections and masking (*e.g.* masking the negation words). Appendix B.2 contains a screenshot and the details of the online evaluation setups.

## 5.4 Results

Table 1 shows the objection evaluation results for both factual and counterfactual rationales on Amazon reviews. Constrained to highlighting 20% of the inputs, CAR consistently surpasses the other two baselines in the factual case for both domains. Compared to the POST-EXP, our method generates the counterfactual rationales with higher precision. However, since the sparsity constraint regularizes both factual and counterfactual generations and the model selection is conducted on factual sparsity only, we cannot control counterfactual sparsity among different algorithms. POST-EXP tends to highlight much more text, resulting in higher recall and F1 score. However, as will be seen later, the human evaluators still favor the counterfactual rationales generated by our algorithm.

Since the beer and hotel datasets contain factual annotations only, we report objective evaluation results for the factual rationales in table 2. CAR achieves the best performances in five out of the six cases in the multi-aspect setting. Specifically, for the hotel review, CAR achieves the best performance almost in all three aspects. Similarly, CAR delivers the best performance for the appearance and palate aspects of the beer review dataset, but fails on the aroma aspect. One possible reason for the failure is that compared to the other aspects, the aroma reviews often have annotated ground truth containing mixed sentiments. Therefore, CAR has low recalls of these annotated ground truth even when it successfully selects all the correct class-wise rationales. Also to fulfill the sparsity constraint, sometimes CAR has to select irrelevant aspect words with the desired sentiment, which decreases the precision. Illustrative examples of the described case can be found in appendix B.3. Please note that the RNP is not directly comparable to the results in [19], because the labels are binarized under our experiment setting.

We visualize the generated rationales on the appearance aspect of beer reviews in figure 3. More examples of other datasets can be found in appendix B.3. We observe that the CAR model is able to produce meaningful justifications for both factual and counterfactual labels. The factual generator picks "*two inches of frothy light brown head with excellent retention*" while the counterfactual one picks "*really light body like water*". By reading these selected texts alone, humans will easily predict a positive sentiment for the first case and be tricked for the counterfactual case.

At last, we present the subjective evaluations in figure 4. Similar to the observations in the objective studies, CAR achieves the best performances in almost all cases with two exceptions. The first one is the aroma aspect of the beer reviews, of which we have discussed the potential causes already. The

poured into pint glass . a : used motor **oil** **color** **.** **two** **inches** **of** **frothy** **light** **brown** **head** **with** **excellent** **retention** **and** **quite** **a** **bit** of lacing . nice cascade going for a while . s : oatmeal is the biggest component of the aroma . not any hops content . a bit fusely and a bit of alcohol . t : tastes like slightly sour nothing . i do n't know **what** **the** hell made this dark because their is no crystal malt or roasted barley component in the taste . this sucks . m : **light** **body** **,** **really** **light** **body** **like** **water** **.** **carbonation** is fine , but that 's about it . d : this is slightly sour water . how does anybody like this ?

Figure 3: Examples of CAR generated rationales on the appearance aspect of the beer reviews. All selected words are **bold and underlined**. Factual generation uses **blue highlight** while the counterfactual uses **red one**.

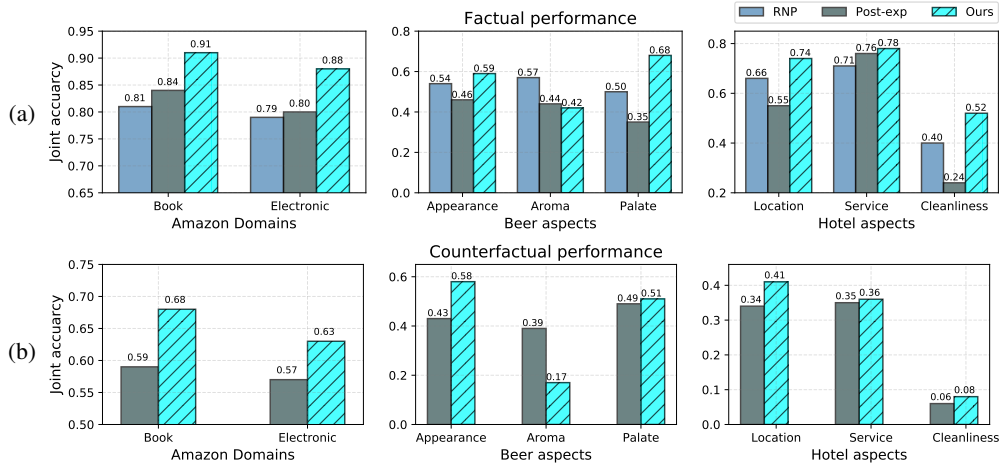

Figure 4: Subjective performances of generated rationales for both (a) factual and (b) counterfactual cases. For the Amazon reviews, subjects are asked to guess the sentiment based on the generated rationales, which random guess will have $50\%$ accuracy. For multi-aspect beer and hotel reviews, subjects need to guess both the sentiment and what aspect the rationale is about, which makes random guess only $16.67\%$.

second one is the counterfactual performance on the cleanliness aspect of the hotel reviews, where both POST-EXP and CAR fail to trick human. One potential reason is that the reviews on cleanliness is often very short and the valence is very clear without a mix of sentiments. Thus, it is very challenging to generate counterfactual rationales to trick a human. This can be verified by the analysis in appendix B.3. Specifically, according to figure 4, $69\%$ of the time CAR is able to trick people to guess the counterfactual sentiment, but often with the rationales extracted from the other aspects.

## 6   Conclusion

In this paper, we propose a game theoretic approach to class-wise rationalization, where the method is trained to generate supporting evidence for any given label. The framework consists of three types of players, which competitively select text spans for both factual and counterfactual scenarios. We theoretically demonstrate the proposed game theoretic framework drives the solution towards meaningful rationalizations in a simplified case. Extensive objective and subjective evaluations on both single- and multi-aspect sentiment classification datasets demonstrate that CAR performs favorably against existing algorithms in terms of both factual and counterfactual rationale generations.

## Acknowledgment

We would like to thank Yujia Bao, Yujie Qian, and Jiang Guo from the MIT NLP group for their insightful discussions. We also want to thank Prof. Regina Barzilay for her support and help.

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
