[Supplementary Material · camera_ready_final_appendix.pdf]

# A  Further Theoretical Discussions and Proofs

In this section, we further our discussion in section 3.4.

## A.1  Proof of Theorem 1

We will now formally prove theorem 1. First, we will briefly state the following lemma regarding the discriminator's optimal strategy.

**Lemma 1.1.** *The global optimizer of Eq.* (1) *is given by*

$$d_0(\boldsymbol{z}) = \frac{p_{\boldsymbol{Z}_0^f,Y}(\boldsymbol{z},0)}{p_{\boldsymbol{Z}_0^f,Y}(\boldsymbol{z},0) + p_{\boldsymbol{Z}_0^c,Y}(\boldsymbol{z},1)}. \tag{11}$$

The proof is similar to [14] and is omitted here.

Theorem 1 is divided into two parts ((1) and (2)). Here we will separately restate and prove each.

**Theorem 1.1. (Restating theorem 1 part (1))** *Assuming equation* (4)*, and given that equations* (11) *is satisfied, the optimal solution to equation* (2)*(left) is given by equation* (7)*.*

*Proof.*  For notational ease, we denote

$$p_{\boldsymbol{Z}_{0,i}^c|Y}(\boldsymbol{z}_i|1) = q_i(\boldsymbol{z}_i), \quad p_{\boldsymbol{Z}_{0,i}^f|Y}(\boldsymbol{z}_i|0) = p_i(\boldsymbol{z}_i),$$
$$p_{\boldsymbol{X}_i|Y}(\boldsymbol{x}_i|1) = Q_i(\boldsymbol{x}_i), \quad p_{\boldsymbol{X}_i|Y}(\boldsymbol{x}_i|0) = P_i(\boldsymbol{x}_i). \tag{12}$$

This notation will be used throughout the proofs in this appendix.

Under equations (4) (11), the optimization problem in equation (2)(left) can be rewritten as

$$\max_{\{q_i(1)\}} \sum_{\boldsymbol{z}\in\{0,1\}^N} \prod_i q_i(\boldsymbol{z}_i) h_1\left(\frac{p_Y(0)\prod_i p_i(\boldsymbol{z}_i)}{p_Y(0)\prod_i p_i(\boldsymbol{z}_i) + p_Y(1)\prod_i q_i(\boldsymbol{z}_i)}\right),$$
$$\text{s.t.} \quad q_i(1) = 1 - q_i(0) \quad \forall i$$
$$0 \le q_i(1) \le Q_i(1) \quad \forall i. \tag{13}$$

For each integer $j < N$, it can be easily shown that each summand in equation (13) is concave in $q_j(1)$ (from equation (3)), hence the summation is concave. Also, taking the derivative w.r.t $q_j(1)$ yields

$$\sum_{\boldsymbol{z}_{-j}\in\{0,1\}^{N-1}} p_Y(0)\prod_{i\neq j} q_i(\boldsymbol{z}_i)\big\{[h_1(\rho_1(\boldsymbol{z}_{-j})) - h_1'(\rho_1(\boldsymbol{z}_{-j}))\rho_1(\boldsymbol{z}_{-j})(1-\rho_1(\boldsymbol{z}_{-j}))]$$
$$-[h_1(\rho_0(\boldsymbol{z}_{-j})) - h_1'(\rho_0(\boldsymbol{z}_{-j}))\rho_0(\boldsymbol{z}_{-j})(1-\rho_0(\boldsymbol{z}_{-j}))]\big\}. \tag{14}$$

where $\boldsymbol{z}_{-j}$ denote a subvector of $\boldsymbol{z}$ without the $j$-th element, and

$$\rho_1(\boldsymbol{z}_{-j}) = \frac{p_Y(0)p_j(1)\prod_{i\neq j} p_i(\boldsymbol{z}_i)}{p_Y(0)p_j(1)\prod_{i\neq j} p_i(\boldsymbol{z}_i) + p_Y(1)q_j(1)\prod_{i\neq j} q_i(\boldsymbol{z}_i)}$$
$$\rho_0(\boldsymbol{z}_{-j}) = \frac{p_Y(0)(1 - p_j(1))\prod_{i\neq j} p_i(\boldsymbol{z}_i)}{p_Y(0)(1 - p_j(1))\prod_{i\neq j} p_i(\boldsymbol{z}_i) + p_Y(1)(1 - q_j(1))\prod_{i\neq j} q_i(\boldsymbol{z}_i)}. \tag{15}$$

When $q_j(1) = p_j(1)$, we have $\rho_1(\cdot) = \rho_0(\cdot)$, and the derivative is 0. Therefore the constrained maximum is achieved at $\min\{p_j(1), Q_j(1)\}$.

$\square$

Before we prove theorem 1(b), we will study the optimal policy of the factual generator *without* the sparsity constraint (equation (6)), as stated below

**Lemma 1.2. (Optimal factual generation without the sparsity constraint)** *Assuming equation* (4)*, and given that equations* (11) *and* (7) *are satisfied, the optimal solution to equation* (2)*(right) is given by*

$$p_{\boldsymbol{Z}_{0,i}^f|Y}(1|0) = \begin{cases} p_{\boldsymbol{X}_i|Y}(1|0), & \textit{if } p_{\boldsymbol{X}_i|Y}(1|0) > p_{\boldsymbol{X}_i|Y}(1|1) \\ \textit{anything}, & \textit{otherwise} \end{cases}. \tag{16}$$

*Proof.* Under equations (11), the optimization problem in equation (2)(right) can be rewritten as

$$\max_{\{p_i(1)\}} \sum_{\boldsymbol{z} \in \{0,1\}^N} \prod_i p_i(\boldsymbol{z}_i) h_0 \left( \frac{p_Y(0) \prod_i p_i(\boldsymbol{z}_i)}{p_Y(0) \prod_i p_i(\boldsymbol{z}_i) + p_Y(1) \prod_i q_i(\boldsymbol{z}_i)} \right),$$

$$\text{s.t.} \quad p_i(1) = 1 - p_i(0) \quad \forall i$$
$$0 \le p_i(1) \le P_i(1) \quad \forall i \tag{17}$$
$$q_i(1) = \min\{p_i(1), Q_i(1)\} \quad \forall i.$$

It can be easily shown that, after substituting $q_i(1)$ with $\min\{p_i(1), Q_i(1)\}$, $\forall i$, the target function is constant in $p_j(1)$ when $p_j(1) \le Q_j(1)$. When $p_j(1) > Q_j(1)$, the derivative w.r.t. $p_j(1)$ is given by

$$\sum_{\boldsymbol{z}_{-j} \in \{0,1\}^{N-1}} p_Y(0) \prod_{i \ne j} p_i(\boldsymbol{z}_i) \big\{ [h_0(\rho_1(\boldsymbol{z}_{-j})) + h_0'(\rho_1(\boldsymbol{z}_{-j}))\rho_1(\boldsymbol{z}_{-j})(1 - \rho_1(\boldsymbol{z}_{-j}))]$$
$$- [h_0(\rho_0(\boldsymbol{z}_{-j})) + h_0'(\rho_0(\boldsymbol{z}_{-j}))\rho_0(\boldsymbol{z}_{-j})(1 - \rho_0(\boldsymbol{z}_{-j}))] \big\}, \tag{18}$$

where

$$\rho_1(\boldsymbol{z}_{-j}) = \frac{p_Y(0)p_j(1) \prod_{i \ne j} p_i(\boldsymbol{z}_i)}{p_Y(0)p_j(1) \prod_{i \ne j} p_i(\boldsymbol{z}_i) + p_Y(1)Q_j(1) \prod_{i \ne j} Q_i(\boldsymbol{z}_i)}$$
$$\rho_0(\boldsymbol{z}_{-j}) = \frac{p_Y(0)(1 - p_j(1)) \prod_{i \ne j} p_i(\boldsymbol{z}_i)}{p_Y(0)(1 - p_j(1)) \prod_{i \ne j} p_i(\boldsymbol{z}_i) + p_Y(1)(1 - Q_j(1)) \prod_{i \ne j} Q_i(\boldsymbol{z}_i)}. \tag{19}$$

When $p_j(1) = Q_j(1)$, the derivative is 0. Considering the function is convex in $p_j(1)$, it will be monotonically increasing when $p_j(1) > Q_j(1)$.

Therefore, when $P_j(1) \le Q_j(1)$, $p_j(1)$ is indifferent within the constraint of $[0, P_j(1)]$; when $P_j(1) > Q_j(1)$, $p_j(1)$ achieves the maximum at $P_j(1)$. □

Now we can turn our discussion back to the case where the sparsity constraint in equation (6) is imposed. First, it is very easy to notice that when the sparsity constraint is mild, the generator can always drop the dimensions where $p_{\boldsymbol{Z}_{0,i}^f|Y}(1|0) \le p_{\boldsymbol{Z}_{0,i}^f|Y}(1|1)$, which we call insignificant dimensions for now, because their probability can be set to anything without changing the target function values. Therefore, the real nontrivial case is when $\alpha$ is so small that dropping all the insignificant dimensions would not suffice, *i.e.*

$$\alpha < \sum_i p_{\boldsymbol{X}_i|Y}(1|0) \mathbb{1}[p_{\boldsymbol{X}_i|Y}(1|0) > p_{\boldsymbol{X}_i|Y}(1|1)], \tag{20}$$

where $\mathbb{1}[\cdot]$ is the indicator function.

We will make a stronger assumption on $h_0(\cdot)$ in addtion to equation (3): the target in eqaution (17) or equivalently in equation (2)(left) under the constraints in equation (17) is jointly convex in $\{p_i(1), \forall i\}$. It can be shown that $\log(\cdot)$ as used in canonical GAN and the linear function as used by CAR both satisfy this assumption.

Now we are ready to restate and proof theorem 1 part (2).

**Theorem 1.2. (Restating theorem 1 part (2))** *Assuming equation* (4)*, the optimal solution to Eq.* (17) *with the constraint in Eqs.* (6) *and* (20) *takes the following form:*

$$p_{Z_{0i|Y}^{(f)}}(1|0) = \begin{cases} p_{X_i|Y}(1|0) & \text{if } i \in \mathcal{I}^* \\ 0 & \text{if } i \in \mathcal{I}^{*c} \backslash \mathcal{J} \end{cases}, \tag{21}$$

*where* $\backslash$ *denotes set subtraction;* $\mathcal{J}$ *can contain either one element or zero. When* $\mathcal{J}$ *contains zero element,* $\mathcal{I}^*$ *satisfies equation* (8)*.*

*Proof.* Since equation (2)(right) is monotonically increasing functions of $p_i(1)$, $\forall i$, the constraint in equation (6) is always binding. Since Eq. (2)(right) is concave, the maximization problem always yields a corner solution, *i.e.* all but at most one $p_i(1)$'s hit the lower bound 0 or upper bound $P_i(1)$. This is because where there are two $p_i(1)$'s that do not hit either bound, the target function is jointly convex with regard to these two quantities along the binding sparsity constraint line (equation (6) with equality). Moving these two quantities along the binding sparsity constraint line will further increase the target.

Notice that equation (8) is essentially the same as the optimization problem in equation (17), but with the constraint $p_i(1) \in [0, P_i(1)]$, $\forall i$ replaced with $p_i(1) \in \{0, P_i(1)\}$, $\forall i$. This is because the target function in equations (8) and (17) are the same. The only difference is that in equation (8), the variable to optimize over is the index set $\mathcal{I}$. When an index $i$ is selected in $\mathcal{I}$, this is equivalent to setting $p_i(1) = P_i(1)$; when $i$ is not selected, this is equivalent to setting $p_i(1) = 0$.

Since $\{0, P_i(1)\}$ is a subset of $[0, P_i(1)]$, the maximum value in equation (8) will be no greater than the maximum value in equation (17).

If optimal $p_i(1)$s in equation (17) either hit the lower bound 0 or upper bound $P_i(1)$ (which means $\mathcal{J}$ contains no element), it will be a feasible solution to equation (8), and therefore should also be the optimal solution in equation (8). $\qquad\square$

## A.2   Reformulating into $f$-Divergence

It turns out that equation (8) can be well interpreted using $f$-divergence. Define

$$f(x) = xh_0(x) - h_0(1). \tag{22}$$

Then the target function of equation in (8) can be rewritten as equation

$$\mathbb{E}_{\boldsymbol{X} \sim p_{\boldsymbol{X}}(\cdot)} \left[ f \left( \frac{p_{\boldsymbol{X}_{\mathcal{I}}|Y}(\boldsymbol{Z}|0)}{p_{\boldsymbol{X}_{\mathcal{I}}}(\boldsymbol{Z})} \right) \right]. \tag{23}$$

It can be easily shown that when $h_0(x) = \log(x)$ (GAN setting) and $h_0(x) = x$ (CAR setting), $f(x)$ is convex, which satisfies the definition of $f$-divergence.

Therefore, under our toy setting, our proposed rationale generator will pick words that satisfy the following two conditions:

- maximizes the $f$-divergence between $p_{\boldsymbol{X}_{\mathcal{I}}|Y}(\boldsymbol{Z}|0)$ and $p_{\boldsymbol{X}_{\mathcal{I}}}(\boldsymbol{Z})$.
- $p_{\boldsymbol{X}_i|Y}(1|0) > p_{\boldsymbol{X}_i|Y}(1|1)$, *i.e.* occurs more frequent in the factual cases than in the counter-factual cases.

## A.3   Inference with Target Label

In section 4, we have discussed how to use CAR to generate rationales without the target label. In fact, CAR can be applied for rationalization when the prediction label is available. For example, when explaining a black-box model [24], we can regard the black-box prediction as the label. In this case, we can make use of the label $Y$ to select between factual and counterfactual generators. For example, when $Y = 0$, we can use $\boldsymbol{g}_0^f(\cdot)$ to generate class-0 rationale and $\boldsymbol{g}_1^c(\cdot)$ to generate class-1 rationale.

## A.4   Generalization to Multiple Classes

So far we have limit our dicussions to two-class classification problems, but CAR can be easily generalized to multiple classes, *i.e.* $Y \in \mathcal{Y}$ where $\mathcal{Y} = \{1, \cdots, C\}$ and $C$ is any positive integer. In this case, there are $C$ factual generators, $\boldsymbol{g}_t^f(\boldsymbol{X})$, $t \in \mathcal{Y}$, each explaining towards a specific class $t$ when $Y = t$. There are $C$ counterfactual generators, $\boldsymbol{g}_t^c(\boldsymbol{X})$, $t \in \mathcal{Y}$, each explaining towards a specific class $t$ when $Y \neq t$.[3]

There are $C$ discriminators, $d_t(\boldsymbol{X})$, $t \in \mathcal{Y}$, each discriminating between $\boldsymbol{g}_t^f(\boldsymbol{X})$ and $\boldsymbol{g}_t^c(\boldsymbol{X})$. The training objective in equation (1) becomes

$$d_t(\cdot) = \underset{d(\cdot)}{\operatorname{argmin}} -P(Y=t)\mathbb{E}[\log d(\boldsymbol{g}_t^f(\boldsymbol{X}))|Y=t] - P(Y \neq t)\mathbb{E}[\log(1 - d(\boldsymbol{g}_0^c(\boldsymbol{X})))|Y \neq t]. \tag{24}$$

The goal of the factual and counterfactual generators is still to convince the discriminator that they are factual. The training objective in equation (2) becomes

$$\boldsymbol{g}_t^f(\cdot) = \underset{\boldsymbol{g}(\cdot)}{\operatorname{argmax}} \, \mathbb{E}[h^f(d_t(\boldsymbol{g}(\boldsymbol{X})))|Y=t], \quad \boldsymbol{g}_t^c(\cdot) = \underset{\boldsymbol{g}(\cdot)}{\operatorname{argmax}} \, \mathbb{E}[h^c(d_t(\boldsymbol{g}(\boldsymbol{X})))|Y \neq t]. \tag{25}$$

Table 3: Statistics of the datasets used in this paper.

| Datasets | | Train | | Dev | | Annotation | |
|---|---|---|---|---|---|---|---|
| | | # Pos | # Neg | # Pos | # Neg | # Pos | # Neg |
| Amazon (single-aspect) | Book | 10,000 | 10,000 | 2,000 | 2,000 | 73 | 27 |
| | Electronic | 10,000 | 10,000 | 2,000 | 2,000 | 261 | 51 |
| Beer (multi-aspect) | Apperance | 16,890 | 16,890 | 6,627 | 2,103 | 923 | 13 |
| | Aroma | 15,169 | 15,169 | 6,578 | 2,218 | 848 | 29 |
| | Palate | 13,652 | 13,652 | 6,739 | 2,000 | 785 | 20 |
| Hotel (multi-aspect) | Location | 7,236 | 7,236 | 906 | 906 | 104 | 96 |
| | Service | 50,742 | 50,742 | 6,344 | 6,344 | 101 | 98 |
| | Cleanliness | 75,049 | 75,049 | 9,382 | 9,382 | 97 | 99 |

where $h^f(\cdot)$ and $h^c(\cdot)$ are monotocially increasing functions satisfying

$$x h^f\left(\frac{x}{x+a}\right) \text{ is convex in } x, \quad \text{and} \quad x h^c\left(\frac{a}{x+a}\right) \text{ is concave in } x, \quad \forall x, a \in [0, 1]. \tag{26}$$

Providing a theoretical guarantee for this multi-class case will be our future work.

## B  Additional Experiments and Details

### B.1  Datasets

The construction process of the three binary classification datasets we used is described below and some statistics of these datasets are summarized in table 3.

**Amazon review:** The original dataset contains customer reviews for 24 product categories. For each product domain, we filter the reviews that are with the patterns "pros: [...] cons: [...]". The goal is to select reviews that separately state the pros and cons of a product so that we could generate both factual and counterfactual rationales automatically using template matching. Specifically, we consider the first 100 tokens after the words "pros:" and "cons:" as the rationale annotations for the positive or negative prediction, respectively. After the filtering, only the domain of book and electronic have sufficient data for evaluation. Thus, we only include these two domains for our experiments. Since the data for both domains are notoriously large, we randomly select 10,000 examples with ratings of two as negative reviews and 10,000 reviews with ratings of four as positive ones. The reason to use ratings of two and four is that we hope to incorporate the data with both positive and negative aspects in a single review. The validation set contains another 2,000 examples for each rating.

**Beer review:** The beer subset introduced by Lei *et al.* [19] for rationalization contains reviews with ratings (in the scale of [0, 1]) from three aspects: appearance, aroma, and palate. Following the same evaluation protocol [8], we consider a classification setting by treating reviews with ratings $\leq 0.4$ as negative and $\geq 0.6$ as positive. Then we randomly select examples from the original training set to construct a balanced set. In addition, the dataset also includes sentence-level annotations for about 1,000 reviews. Each sentence is annotated with one or multiple aspects label, indicating which aspect this sentence belonging to. We use this set as factual evidence to automatically evaluate the precision of the extracted rationales.

**Hotel review:** The dataset contains reviews of the following three aspects: location, cleanliness, and service. We use the same data provided by Bao *et al.* [8], where reviews with ratings $> 3$ are labeled as positive and those with $< 3$ are labeled as negative. Similarly, for each aspect, the dataset contains 200 examples with human annotations, which explains why a particular rating is given.

### B.2  Experiment settings

The details of our experiment settings are as follows:

**Objective evaluation:** As we mentioned in the main paper, we compare the generated rationales with the human annotations and report the precision, recall and F1 score. For fair comparison, the evaluation is conditioned on a similar *actual* sparsity level in factual rationales (the target sparsity level is set to 10% for both beer and hotel review, and 20% for the Amazon dataset), which requires

Figure 5: A screenshot of the user interface of the subjective evaluation on hotel review.

tuning the hyperparameters $\lambda_1$, $\lambda_2$ and $\alpha$. However, only the annotation set contains a fairly small number of the annotated rationales, it is hard to separate the set or generate more examples for the validation purpose. In order to control the actual sparsity and avoid overfitting the annotation set, we adopt the following setup to determine the hyperparameters. For each method, each hyperparameter is drawn from five candidate values, and we report the best test performance within $\pm 2\%$ of the preset sparsity level on factual rationales. This experiment setting is consistent with the one that has been reported previously [19].

As for determining the number of training steps, that for RNP and the POST-EXP predictor is done by maximizing the sentiment prediction accuracy on the dev set[4], because their training goals are both to maximize the sentiment prediction accuracy. On the other hand, since the dev set does not have rationale annotations, the number of training steps of POST-EXP generators and our method has to be preset, guided by the principle to match what is needed for RNP to achieve a good dev performance, which is around 15 epochs. It is worth mentioning that all algorithms are trained on the training set, except for the POST-EXP generators, which are directly optimized based on the examples we evaluate. This is because the training of the generators does not require any supervision, as is the case for other gradient-based methods for model interpretations.

**Subjective evaluation:** Figure 5 shows the screenshot of an example test in the subjective evaluation on multi-aspect hotel review. As can be seen, the subjects are presented with text with blocks. Only the rationales are revealed. The subjects are asked to guess both the aspect and the sentiment of the aspect. Figure 6 shows the instructions that were presented to the subjects before the test. For the single-aspect datasets (Amazon review), the test is almost the same, except that the subjects do not need to guess the aspect.

### B.3 Additional results

Additional experiment results are shown below:

**Understanding the aroma aspect of beer reviews:** To better understand the lower performance of CAR on the aroma aspect of the beer reviews, we illustrate two examples of generated rationales in figure 7. We see that most of the factual and counterfactual rationales that are both only a subset of the ground truth annotation. This because the annotations contain a mix of sentiments. In the second example, we also see the counterfactual generator selects other text spans to prevent punished by the sparsity constraint.

**Additional illustrative results:** In addition to table 3 in the main paper, we include more illustrative results in table 8 and 9. Particularly, table 8 visualizes the generated rationales on the electronic domain of Amazon reviews while the latter table includes examples from the multi-aspect datasets.

Figure 6: Instructions of the subjective evaluation on hotel review.

Table 4: Subject evaluation results on the beer review dataset. 'Sentiment" and "Aspect" are the marginal performances of the "Joint" results.

| Beer Review | | Appearance | | Aroma | | Palate | |
|---|---|---|---|---|---|---|---|
| | | Factual | Counter | Factual | Counter | Factual | Counter |
| Sentiment | RNP | 0.72 | - | 0.72 | - | 0.61 | - |
| | POST-EXP | **0.86** | 0.64 | 0.77 | **0.64** | 0.70 | 0.57 |
| | CAR | 0.85 | **0.68** | **0.81** | 0.59 | **0.85** | **0.69** |
| Aspect | RNP | 0.72 | - | **0.70** | - | **0.80** | - |
| | POST-EXP | 0.52 | 0.43 | 0.53 | **0.39** | 0.48 | 0.49 |
| | CAR | **0.72** | **0.58** | 0.48 | 0.17 | 0.75 | **0.51** |
| Joint | RNP | 0.54 | - | **0.57** | - | 0.50 | - |
| | POST-EXP | 0.46 | 0.28 | 0.44 | **0.24** | 0.35 | 0.25 |
| | CAR | **0.59** | **0.41** | 0.42 | 0.13 | **0.68** | **0.43** |

Similarly, we observe that CAR produces meaningful justifications for both factual and counterfactual labels.

**Additional subjective evaluations:** Table 4 and 5 illustrate comprehensive results of the subjective evaluations for the beer and hotel reviews, respectively. The results in the bottom section (*i.e.,* "joint") of the tables are included in figure 4 in the main paper. Recall that in the multi-aspect subjective experiments, the subjects need to answer both aspect category and the sentiment valence. Both tables include the accuracy of each marginal statistics of the results, which are shown as "sentiment" and "aspect" in the tables. The former one records sentiment accuracy regardless of whether the aspect is correctly classified while the latter one represents the aspect accuracy only.

*Beer - Aroma*                                                                                     Label - Positive

poured from a bottle into a pint glass . a : pours a ruby amber color with 1/4 ' thin head and some lacing . this is a good looking **beer . s : floral hops and citrus burst into the nostrils . some slight malt and pine aromas also present** . t : sweet citrus flavors dominate the initial sip followed by a significant hop presence in the middle to finish . some slightly unappealing bitter aftertaste in the finish . m : this beer has the feel of a good spring beer . it is light and thin on the tongue with the presence of prickly carbonation . overall this is a good beer . it is very similar in flavor to nugget nectar , but the flavors and character are not as prominent as in its imperial amber ale cousin . aside from the slightly bitter aftertaste , the beer is well balanced and one of the tastier amber ales that i have had .

*Beer - Aroma*                                                                                     Label - Positive

small somewhat creamy light brown head that mostly diminished . **pitch black color . aroma of bread , molasses , caramel , chocolate , coffee** , toasted **malt . light alcoholic sniff** , plum , prune , date , licorice . very complex aroma . full bodied with an alcoholic texture and a soft carbonation . excellent balance between **the moderated sweet malt** and bitter flavor . berries sweet and lingering finish . smooth and very nice thick beer . definitely one of my favorite ris .

Figure 7: Examples of CAR generated rationales on the aroma aspect of the beer reviews. The provided ground truth annotations are shown in the blue text. All rationale words are **bold and underlined**. To distinguish the factual and counterfactual generation, we use **blue highlight** to represent the former one while **red highlight** for the latter one.

*Amazon - Electronic*                                                                              Label - Positive

**these are good , solid headphones . i ca n't say they blew me away , <unknown> , the frequency response is very flat** . with & # 34 ; street by 50 cent & # 34 ; on the box , i was kind of expecting them to be seriously bass-heavy , but that 's not the case . **i listened to quite a variety of different music , and was pleased with their performance across the board.i hesitate to refer** to & # 34 ; noise canceling , & # 34 ; since there is no active nc going on here , but the ear cups do a decent job of blocking out outside sounds . they are soft and have smooth covers , and i can go about an hour **before they start to get uncomfortable . not bad** , but could be better.the star wars themed design and accessories are pretty cool . this <unknown> fett model will only be recognizable to the hardcore fan , but the design is nice looking anyway .

*Amazon - Electronic*                                                                              Label - Negative

**i purchased the edimax <unknown> to take advantage of** the 5ghz band from my asus dark knight router , **which is an awesome** router by the way . **installation from the cd was easy** and then updating the firmware using a download from the edimax website was **also easy .** however , **i was really disappointed in the performance .** i used inssider 3 to verify connection to the 2.4 or 5 ghz band . 95 % of the time the edimax would connect to the 2.4ghz band and not the 5ghz band even though the 5ghz band was showing to be present via inssider <unknown> when it was connected to the 5ghz band the best speed was <unknown> download , which is **not too bad** . yet most of the time , the edimax connected to the 2.4ghz with download speeds of <unknown> or less . the strangest thing i noticed about the edimax adapter was that is did not sustain a consistent signal from the router only 8 feet away . during the ookla download tests , the signal always had a saw tooth appearance and **the speed needle swung <unknown> 'm going to put the tp** link <unknown> **back on the laptop which consistently beats** the edimax adapter . the tp link download speeds were 38 to 55mbs on the 2.4ghz <unknown> to put these numbers into perspective , i have fios <unknown> and get wired download speeds of about <unknown> and wired uploads of <unknown> .

Figure 8: Examples of CAR generated rationales on the electronic domain of the Amazon reviews. All selected words are **bold and underlined**. Factual generation uses **blue highlight** while the counterfactual uses **red one**.

Table 5: Subject evaluation results on the hotel review dataset. "Sentiment" and "Aspect" are the marginal performances of the "Joint" results.

| Hotel Review | | Location | | Service | | Cleanliness | |
|---|---|---|---|---|---|---|---|
| | | Factual | Counter | Factual | Counter | Factual | Counter |
| Sentiment | RNP | 0.78 | - | 0.87 | - | 0.88 | - |
| | POST-EXP | **0.84** | 0.53 | 0.89 | 0.55 | 0.84 | 0.39 |
| | CAR | 0.83 | **0.53** | **0.93** | **0.56** | **0.96** | **0.62** |
| Aspect | RNP | 0.83 | - | 0.81 | - | 0.42 | - |
| | POST-EXP | 0.65 | 0.62 | 0.81 | 0.58 | 0.27 | **0.15** |
| | CAR | **0.87** | **0.77** | **0.84** | **0.62** | **0.52** | 0.13 |
| Joint | RNP | 0.66 | - | 0.71 | - | 0.40 | - |
| | POST-EXP | 0.55 | 0.34 | 0.76 | 0.35 | 0.24 | 0.06 |
| | CAR | **0.74** | **0.41** | **0.78** | **0.36** | **0.52** | **0.08** |

---

*Beer - Appearance*                                                            Label - Positive

poured into pint glass . a : used motor **oil color . two inches of frothy light brown head with excellent retention and quite a bit** of lacing . nice cascade going for a while . s : oatmeal is the biggest component of the aroma . not any hops content . a bit fusely and a bit of alcohol . t : tastes like slightly sour nothing . i do n't know **what the** hell made this dark because their is no crystal malt or roasted barley component in the taste . this sucks . m : **light body , really light body like water . carbonation** is fine , but that 's about it . d : this is slightly sour water . it fucking sucks . how the hell does anybody like this ?

---

*Beer - Appearance*                                                            Label - Negative

got this at de bierkoning in amsterdam . from a bottle into a mug . appearance : pours **a very small and thin off-white head that quickly disappears** . lower level carbonation evident , which quickly ceases as well . colour is more brown than red , like a sienna with some faint hints of rust on the sides . **zero film stays and** no lacing . relatively lack **luster** , even for the style . smell : a medium strong nose of sweet caramel malt , a bit of toffee and maybe some red fruit in the back . a little one dimensional , but nice enough . taste : sweet caramel malt in the front with some toffee as well , and finishes with a burst of spicy alcohol at the end . aftertaste **is nice , mild and long lasting** , consisting of some spicy alcohol and a slight touch of bitterness . not bad ; more impressive than the nose . palate : medium body and medium carbonation . **relatively creamy on** the palate , goes down quite smooth with a small alcohol bite at the very end . finishes on the stickier side of the spectrum . overall : not bad , but not memorable . the look was certainly disappointing , but it was pretty average other than that . drinkable enough , but not worth seeking out .

---

*Hotel - Service*                                                              Label - Positive

i would definitely recommend this hotel to anyone who has plans at the staples center or nokia theater **. the staff was friendly and helpful , the rooms were clean , and there was plenty of entertainment and dining options within walking distance . they provide robes , slippers , and** wonderful toiletries ! the shower was beautiful but **the head was not working properly and we requested a non smoking room but there were n't any available . the smell of stinky cigarettes really disgusted me** but if they had a non smoking room available i honestly would not have any complaints . i recommend taking a car to and from the airport . i would stay at this hotel again for sure .

---

*Hotel - Service*                                                              Label - Negative

**i loved the hotel and the hotel room . my daughter and i were very impressed .** we had a lovely room and for being a hilton honors **member we even received a free fruit basket** and two bottles of water . please beware however and always check your bank account for any additional charges . we had used my daughters credit card to hold the room but asked for my visa debit card to be charged for the final amount . we were both charged for the room **and it took numerous phone calls to receive a credit . we were also charged for several additional charges that we** should not have been . the billing department was extremely rude as we called them numerous times to get the matter straightened out . as for the hotel i would **highly recommend it and also for it 's wonderful** location and shuttle . i just did n't appreciate the cutomer service afterwards and the fact that it took me so long to get the matter taken care of . just be very careful looking over your final billing .

Figure 9: Examples of CAR generated rationales on the multi-aspect datasets. All selected words are **bold and underlined**. Factual generation uses blue highlight while the counterfactual uses red one.

## Footnotes

[3]There is no further differentiation of the counterfactual generators. In other words, the game is still played by three players. The counterfactual generator of each class, no matter what the ground truth label is, is considered as only one player.

[4]Although the dev set does not have rationale labels, it has sentiment labels.