[Reviews · NeurIPS 2019]

Reviewer 1



The paper studies an important problem, that is to learn models to localize "rationales" from a corpus of polarized opinions (such as product reviews), all in an "unsupervised" fashion (i.e. without the presence of ground truth rationales). Different from a previous pioneering work [18], this paper tries to further pin down rationales that are class-specific, so that both "pros" and "cons" would be simultaneously identified. To me this is a meaningful extension and the paper is largely well written and easy to follow. Detailed comments as follows, 1. I wonder if the authors have tried to learn class-specific rationales that are ground-truth *agnostic*. In a potentially simplified setting, you can still have class-specific rational generators (e.g. one for localizing "pro" rationales and another for "con"), but they do not necessarily need to be further tied with ground-truths so as to differentiate between "factual" and "counterfactual" (more on this later in comment 4)? 2. L.183: "For the discriminators, the outputs of all the times are max-pooled ..." - why choose max-pooling-over-time rather than a simple sequence classifier that directly outputs 0/1? 3. Eq.10: cite [18] since this is essentially the same regularizer introduced there? 4. For the baselines, if we adopt the simpler setting as outlined earlier in comment 1, it would be interesting to consider another model that's basically RNP with 2 "class-specific" generators that share parameters and take the class-label as an additional input. It's closely related to POST-EXP yet will benefit from a jointly trained predictor? 5. Why are the sparsity levels in Table 1 and 2 not exactly the same across the three models in comparison? Assuming the three models all make "independent" selection decisions (per L.179), it should be straightforward to enforce a common exact number of input tokens to keep, by selecting the top-K positions? 6. Comparing Table 2 with Table 1, the performance of POST-EXP sees a drastic drop, from consistently outperforming RNP to underperforming. Why is that? === post author response === Thanks for your clarifications. I believe having some of them in the paper will help your readers appreciate it more and clear away similar confusions. That said, I'm still not quite convinced why a class-specific yet ground-truth-agnostic RNP extension would yield degenerate results - are you suggesting the classification task per se encourages the model to exploit "spurious statistical cues" in the dataset more than the factural vs. counterfactural classification task?

Reviewer 2



ORIGINALITY & CLARITY: The paper picks out interesting direction and clearly motivates the idea in the introduction. The figures are helpful to understand the paper. QUALITY & SIGNIFICANCE: The main concern that I have with this paper is that it doesn't have head-to-head comparison with the existing literature, notably Lei et al EMNLP 16. Ideally, they should have performed evaluation that could be compared to the previous paper. From what I understand, Table 2 from EMNLP 16 paper could have been compared to Table 2 in this manuscript. I'm not convinced why the numbers are so different here. The subjective evaluation should have done more thoroughly. What was inter annotator agreement between different crowd workers? Given such a small sample size (100), strong agreement is necessary for the results to be meaningful. ==== AFTER THE AUTHOR RESPONSE: Thank you for clarification.

Reviewer 3



Specifically, by setting the architecture as an adversarial network the idea is that the predictor (in this case the discriminator) is trained to not only see good explanations, it is also exposed to counterfactual examples and can better tell the difference. This is a valuable and original use of adversarial models in the area of transparency. This submission is a game theoretic submission but the game itself is a very simple one so I would suggest that this paper be classified more under transparency as that is where it has the biggest contribution to current literature. Additionally while the 3-player game was very well modeled and explained at the end the implementation was reduced to a 2-player game with a generator and a discriminator. It would have helped if the authors had made this explicit and simplified their formulation. That said I would say this is a great contribution to the transparency literature and I enjoyed reading this submission.

[Author Response · NeurIPS 2019]

We thank all the reviewers. Based on the valuable comments, we provide the following rebuttal to address the concerns of each reviewer, respectively.

**Response to Reviewer 1**    Thanks for your constructive feedbacks! Regarding your comments:

*Comments 1 & 4*: It turns out that RNP (The baseline in "Rationalizing Nueral Predictions" proposed by Lei et. al.) with class-specific generators will converge to a set of degenerated solutions, where, rather than highlighting the informative words for each class, the generators will communicate the class information using trivial symbols (*e.g.* punctuation) or other uninformative features (*e.g.* word position). In one degeneration example, the class-0 generator will always highlight the first word and the class-1 generator will always highlight the last word. As a result, the predictor always has 100% prediction accuracy, which achieves the global optimum of the collaborative target function of RNP. Our empirical experiments verify that RNP with class-specific generators quickly converges to degeneration.

In fact, even the original RNP suffers from the degeneration problem. The additional class input merely deteriorates the problem. The problem primarily results from the collaborative nature of the RNP framework. Furthermore, we are able to show that the adversarial game in CAR can fundamentally resolve the degeneration problem. This is because any attempt to communicate class information using uninformative symbols will be easily mimicked by the counterfactual generator, and therefore would not be a good strategy for the factual generator, whose goal is to prevent mimicking. This is another major advantage of CAR, which we did not have enough space to uncover in the paper. Nevertheless, we will add the above discussion to the paper.

*Comment 2*: Max-pooling followed by a feed-forward classifier is a commonly-used and well-performed sequence classifier in the sentiment classification community (*e.g.,* Yoon Kim, EMNLP 2014). Therefore we simply follow the usage. Moreover, in our experiments when GRUs are used, max-pooling makes the training converge faster compared to using the final hidden state, which is beneficial for the game training.

*Comment 3*: Eq. (10) is slightly different from [18] in the first term as we hope to better control of the lengths for apple-to-apple comparison. We will add the reference in the updated version.

*Comment 5*: Directly highlighting top-$K$ positions during the testing phase can be undesirable because during training we used Eq. (10) to constrain on the *average* sparsity level, not the *per-passage* sparsity level. The theoretic guarantee of CAR is also contingent on the average sparsity level. Per-passage sparsity stipulates that each passage has exactly the same proportion of rationale, which is less theoretically and empirically sound, and which also creates a training-test discrepancy if applied to testing only. Using top-$K$ projection for training is not desirable either because the top-$K$ ball is non-convex and does not lead to good convergence.

*Comment 6*: RNP and POST-EXP come with their respective strengths and weaknesses. RNP is good at finding the aspects that correlate most closely with the output labels, but lacks class-specific capabilities; whereas POST-EXP is good at finding class-specific words and phrases, but does poorly in finding the right aspect. That is why the two baselines perform differently in Tables 1 and 2. Amazon reviews (Table 1) mostly contain single aspect, but has a lot of mixed reviews, and thus POST-EXP works better. Beer and hotel reviews (Table 2) contain multiple aspects, but within each aspect, the reviews are less mixed, and thus RNP performs better. This can be verified by Tables 4 and 5.

**Response to Reviewer 2**    We greatly appreciate your acknowledgment of the paper. Regarding your concerns:

*Direct comparison to Lei et. al. paper*: The experiment in the Lei et. al. paper is a regression task, where the output scores are real values within $[0, 1]$. In our experiment, we convert the task to a classification task (because class-wise rationales are only well-defined on classification tasks so far) by quantizing the real-valued scores into binary scores. The quantization operation squeezes out information, and deteriorates the performance of the RNP baseline. We explained this difference in the paper (line 213), but we will emphasize it more in the updated version.

*Inter-annotator agreement*: Each HIT was originally assigned to one crowd worker. Therefore, we rerun the experiment and obtain a second set of scores. We then compute the agreement between the two sets of scores. Please note that despite that the agreement score can be high or not so high, the new set of accuracy is consistent with the one reported in the paper. For our method on factual rationales, the sentiment agreement is 0.71 on beer reviews, 0.79 on hotel reviews and 0.81 on Amazon reviews. The aspect agreement is 0.58 on beer reviews and 0.64 on hotel reviews. The aspect agreement is lower because for many cases people are making random guesses among the three options, which can be confirmed by the low accuracy in some aspects (beer aroma and hotel cleanliness) in Tables 4 and 5.

**Response to Reviewer 4**    We greatly appreciate your positive feedback! Please note that although there is only one generator network for each class in our implementation, there are still two generator players for each class. The two players are differentiated by feeding the ground truth label as an additional input to the generator network, and they still have completely distinct target functions in the game. We will make this point clearer in our updated version.

[Meta-Review · NeurIPS 2019]

A solid paper with on an interesting and increasingly important problem: finding rationalizations/explanations for model predictions (in a sentiment classification task). The reviewers are all positive and after some discussion also fairly confident in their assessments.